# Biolaminates as an Example of Upcycling Product with Keratin Flour—Research and Thermal Properties Modeling

**DOI:** 10.3390/ma17164081

**Published:** 2024-08-16

**Authors:** Michał Frydrysiak

**Affiliations:** Faculty of Materials Technology and Textile Design, Textile Institute, Lodz University of Technology, 90-924 Łódź, Poland; michal.frydrysiak@p.lodz.pl

**Keywords:** keratin flour, upcycling materials, textiles, biolaminates, thermal insulation, modeling

## Abstract

Keratin waste, including keratin powder, is a significant byproduct of the poultry processing and meat industries. It is a major contributor to waste management problems due to its volume and the environmental pollutants that it can produce. The disposal of keratin waste is challenging due to the potential for odors and pathogens to enter the soil and water. The aim of this work is to present the possibility of using waste materials in accordance with the principles of upcycling and producing fully valuable products. In this research, the author focuses on the production and research of textile multilayer laminates using keratin flour that had been previously considered waste material. New textile composites should be characterized by increased thermal insulation properties with constant comfort in use. This research determines the physiological comfort interpreted as the state of the human–laminate system, which maintains the conditions of comfort in human perception, i.e., constant temperature and humidity of the body under changing conditions of a relative humidity environment.

## 1. Introduction

Keratin powder waste, primarily sourced from the poultry, meat, and wool industries, is a significant environmental challenge. The management of this waste is crucial due to its potential to release odors, pathogens, and pollutants into the environment. Keratin waste is classified under specific regulations that concern animal byproducts not intended for human consumption. The management of this waste must consider environmental risks, especially in the poultry farming industry and landfills. The use of keratinolytic microorganisms offers an ecological solution by effectively degrading keratin waste and reducing its environmental impact [1].

Feathers, in particular poultry feathers, are produced during the slaughter of poultry and constitute from 4 to 7% of their live weight. Their further processing is carried out at utilization factories, which transforms them into poultry meal [2]. Before 2004, this meal was added to industrial feed. However, the European Parliament has banned its use in feeding farm animals. Currently, flour can only be used as an organic fertilizer or an additive for combustion in industrial boiler houses. Nowadays, biomaterials are created with feather keratin because chicken feathers consist of 95–96% keratin—a protein with many interesting and specific properties that allow its wide use. Therefore, keratin powder can be used, e.g., as a drug carrier or paper-like material [3]. The article [4] describes the threats resulting from the constant increase in poultry production. Feather waste is a byproduct generated on a massive scale by the poultry industry globally. Bird feathers are composed primarily of keratin, which is an insoluble, fibrous protein that constitutes 85% to 99% of the total dry weight of feathers.

However, feather waste is also a serious problem that can lead to environmental pollution, as many pathogens and microorganisms such as *Salmonella* and *Vibrio* can be found in poultry feathers. Various pollutants such as ammonia, nitrous oxide, and hydrogen sulfide can also be emitted from feather waste, which may cause hazards and risks to both human health and environmental safety [4,5].

The future of keratin waste management lies in developing cost-effective, time-efficient, and environmentally friendly methods for its extraction. Further research is needed to understand the structure–activity relationship during the preparation of keratin-based biofilms and to explore new applications of these materials. Keratin powder waste management is evolving with advances in biodegradation techniques and the discovery of new uses for extracted keratin. Sustainable practices not only reduce environmental risks but also add value to waste biomass, contributing to a circular economy. Using bird feathers as a raw material for biolaminates could contribute to sustainable development by reusing natural resources that might otherwise be wasted. This is consistent with the principles of a circular economy and resource efficiency. Further research and development is necessary to fully exploit the potential of keratin waste as a resource for various industries. Therefore, this article presents the potential use of keratin flour as an insulating material.

Since the dawn of time, people have protected their bodies from weather conditions, especially low temperatures. Analyzing history, it can be observed how people have developed clothing, starting from the first clothes made of animal skins through advanced technological solutions [1]. Today, clothing is basic everyday equipment that fulfills various functions in our society and at the same time protects our lives and health from unfavorable environmental conditions [6].

For many years, scientists have been working on creating materials that are characterized by good thermal insulation properties and, at the same time, high comfort of use. Materials are made from various types of raw materials, ranging from natural materials such as cotton, wool, and various types of natural fibers such as linen or hemp, to synthetic raw materials such as polyamide, polyester, carbon fibers, etc. [7,8,9]. This complex and multithreaded issue can be analyzed from various points of view [9].

In this work, the author uses bird feathers in the form of flour to construct laminates. This material is widely used in industrial production, e.g., for warm jackets in the form of traditional feathers or as a component of feed in the form of flour. New environmental and economic aspects, as well as the depletion of oil resources, have led the scientific community to take greater interest in the problems of waste recycling, including natural waste materials [10]. The essence of this article is the selection of component materials for the multilayer textile laminate in such a way that the designed system would ensure, within the range of usage variability assumed by the author, that the values of thermal resistance and specific conductivity coefficients could be maintained at an unchanged level.

## 2. Materials and Methods

In this study, several types of layered materials were designed and tested. The aim of this work was to use waste material generated during the production of chicken feathers, namely, ground bird feathers waste in the form of keratin flour. Feather hydrolysate is produced in the process of thermal hydrolysis of feathers under the pressure of 5 bar to split keratin bonds. The technology used in this process allows us to obtain a product with high protein digestibility. The proportion of blood and feathers is analogous to their proportion in living birds. The powder contains 100% poultry substance. The raw material/stock for production comes from healthy animals (mixed breeding—poultry, turkey and geese), the meat of which has been recognized by the Veterinary Inspection as suitable for human consumption. The powder is stabilized with positive antioxidants BHA and BHT. It does not contain ethoxyquin (E324) or any other antioxidant [11,12].

Similar work has been conducted by other researchers in this field [2,13]. However, in the cited works, the authors added flour to thermoplastic materials, thus creating a kind of biocomposite polymer matrix. Other works were also carried out on the introduction of keratin flour directly into the structures of nonwoven fabrics (in the structure of thermoplastic fibers), but did not include the modification of the nonwoven fabrics themselves and did not include testing of the thermal properties of the material [14,15].

In the presented article, the laminate consisted of a hydrophilic nonwoven fabric on the underside. The other side was covered with keratin flour. Various variants of the mass fractions of meal inputs in the package were tested.

The aim of the research was to determine the influence of the relative humidity of the environment and the mass of moisture supplied to the bottom layer of the biolaminate on the values of its thermal resistance and thermal conductivity.

According to the author, such a combination of selected materials in the composite is characterized by higher thermal comfort, tested in the conditions of variable relative humidity and at different levels of moisture transport through the package, which may occur in the case of human sweating [16].

The main assumption in the construction of the biolaminate was the use of keratin flour, which is made from waste generated during the production of chicken feathers. The meal obtained from feathers after hydrolysis was a loose, homogeneous powder, without charred particles and bodies other than poultry feathers. It had a specific odor and a dark brown color.

Previously, bird feathers, as well as claws and beaks, were processed into meal or fodder, but now this is banned in the European Union. Therefore, only processed feathers were used for the research. This material was supplied by one of the producers on the Polish market. The philosophy of upcycling in the production economy is currently very fashionable and rational, which is why the author was looking for new uses for waste products such as ground bird feathers. Example photos of the source material and keratin flour are shown in Figure 1. The producer of the flour was the Polish company Kemos from Bialystok [11].

A single grain is shown in Figure 1C,D. In order to characterize the research material, measurements of the grain diameter of keratin flour in the batch were performed with the use of the microscopic method. In the study, the dimensions of the flour particles were determined using the equivalent diameter.

The measurement was made for the flour subjected to the process of acclimatization and stabilization under normal climate conditions. The test samples were acclimatized in a Radwag MA 50.R weighing dryer (Radwag, Radom, Poland)with an elementary scale of 0.001 g and a measurement accuracy of 0.01%. The results of the diameters were tested for the raw material moisture content of 7%. In order to measure the grain size, microscopic photos of keratin flour granules were taken. The granulate was characterized by an irregular shape; therefore, in addition to the measured values of equivalent diameters, the coefficient of variation of the granulate diameter was given, which was 149.83 µm, and the coefficient of variation was 16.4%. For this coefficient, when its value is less than 25%, then variability of the diameter dimensions of the tested sample is small. That means that there is uniformity of keratin flour grains in the batch. Another component of the laminates is textile material selected by the author in the form of polyester nonwovens subjected to the hydrophobization process. A satisfactory level of hydrophobization of the polyester nonwoven fabric was obtained by modifying its surface with hydrophobic agents. It was padded in an aqueous dispersion of dodecyldimethylamine at a temperature 150 °C. Based on the research carried out, it was assumed that the concentration of 40 mL per 1 L of water was beneficial. The commercially available Kemiline Guard YSA from Kemitekst (Istanbul, Turkey) was used as a hydrophilizing formulation. The keratin flour was bonded to the surface of textiles with the use of a TensorGrip L14 adhesive contact spray (Tensor, St. Neots, UK), which has a negligible heat resistance. Taking into account the design assumptions of the laminate and the materials described above, the subject of the research was laminates made of a top layer of polyester nonwoven with a hydrophobic coating and, optionally, a bottom layer with keratin powder with four variants of backfill density. The diagram of the laminate variants and the measurement model are shown in Figure 2. The structure and thickness of the laminates covered by the research are summarized in Table 1. The coefficient of variation for the area weight was ±4 g, and for the thickness it was ±0.01 m. Thermal insulation properties were measured [14,17,18,19,20].

Thermal tests were performed in accordance with the standard [21]: based on thermally insulated sweating plate method). The tests of the thermal insulation properties of the laminate—thermal resistance and thermal conductivity coefficients were carried out for two levels of relative humidity: 40% and 80%, and two values of the moisture mass delivered to the bottom layer of the laminate in half an hour: 0 and 1.5 mL. This value was selected after calculating the tested surface area so that it corresponded to the amount of sweat that a person loses under normal conditions, without effort and during physical exertion [22]. Hence, during the tests, the initial value was assumed to be 1.5 mL of sweat and moisture introduced into the gap between the surface of the emission of moisture and the surface of the laminate.

### 2.1. Measurement Methodology

The aim of the test was to determine the thermal insulation properties of laminates made of keratin flour. These studies were the basis for later work on defining physiological comfort, understood as the state of the human–laminate system. It maintained human-perceived comfort conditions, i.e., constant body temperature and humidity under conditions of relative humidity and with increased sweat secretion from the human body into the textile composite. The criteria for selecting the optimal laminate structure were as follows:The ability of the laminate to reduce heat emission from the surface of the human body for various variants of the biocomposite;The extent of the decrease in the thermal insulation properties of the laminate with the assumed range of increase in the relative humidity of the environment;The extent of the decrease in thermal insulation properties of the laminate with the assumed range of increase in the transmission of sweat vapor from the surface of the human body to the laminate substrate.

The optimal structure of the laminate was characterized by the smallest decrease in the thermal insulation properties of the laminate for the assumed increase in the ambient relative humidity of the environment and the mass of moisture supplied to the bottom layer of the laminate. The following were adopted as measures of the physiological comfort of the laminate:Thermal resistance coefficient;Thermal conductivity coefficient.

The measurements of thermal conductivity and thermal resistance coefficients were performed for relative air humidity of 40% and 80% at a temperature of 21 °C. The tests were based on [21] the determination of physiological properties, measurement of thermal resistance, and water vapor resistance under steady state conditions. The methodology for calculating the thermal conductivity coefficient was based on the following dependencies, and the heat flowing through the partition q is determined by Formula (1):q = U·S·ΔT [W](1)
where

U—heat transfer coefficient;

q—amount of heat flowing per time unit (heat flux);

S—surface area;

ΔT—temperature difference between both sides of the partition.

The heat transfer coefficient U was determined by the Formula (2):U = λ/D [W·m^−2^ K^−1^](2)
where

λ—coefficient of thermal conductivity;

D—the thickness of the laminates.

The heat transfer coefficient (U) is a coefficient that determines the amount of heat penetrating through the biolaminate materials. This coefficient depends on the surface area and the temperature difference on both sides of the materials. This coefficient is important when assessing the thermal insulation of materials.

Therefore, the thermal conductivity coefficient λ was calculated according to Formula (3):λ = U·D [W·m^−1^·K^−1^](3)

Based on Formula (1), the formula for the heat transfer coefficient U was
U = q/S·ΔT(4)

Taking into account the above formula in calculating the thermal conductivity coefficient (4), this value was determined by Formula (5):λ = D·q/S·ΔT [W·m^−1^·K^−1^](5)

### 2.2. Apparatus Used in Research

(1)A climatic chamber stabilizing the humidity and temperature parameters;(2)Matrix laboratory power supply model MPS-3003L-3 (Matrix Orbital, Calgary, AB, Canada);(3)Insulated hob with a total power of 12 W; heating area 0.00847 m^2^ (LUT, Lodz, Poland);(4)Amprobe TMD90A thermometers (0.1% rdg + 0.5 °C) (Amprobe, Washington, DC, USA);(5)AXIO MET AX 594 multimeter (Axiomet, Malmö, Poland);(6)Moisture Meter, range 0~100% RH; accuracy ≤ ±4% RH (Axiomet, Malmö, Poland);(7)Skin Model, PN-EN 31092:1998/A1:2013-03E [21] (LUT, Lodz, Poland);(8)Infrared camera, Flir SC 5000, 320 × 256 pixel resolution, 0.02 K sensitivity, 150 Hz acquisition frequency (Flir, Portland, OR, USA).

## 3. Results and Discussion

### 3.1. Experimental Results

The tests results of thermal conductivity coefficients and thermal resistance of laminates under assessment are summarized in Table 2 and in Figure 3 and Figure 4.

Thermal analysis of the biolaminates photos presented in Figure 5 shows the temperature distribution on their surface. For that analysis, the Flir SC 5000 infrared camera (Flir, Portland, OR, USA) was used. Based on this, we can observe the uneven application of keratin flour. The standard deviation coefficient of temperatures in the tested area can be a unit of measurement of the unevenness of flour application. The smaller the deviation value of the flour used, the more the observations are concentrated around the mean value, which allows us to conclude that the flour distribution on the laminate surface is more uniform. Variant V4 is the one in which the uniformity of flour application is the greatest, and the quantitative share of keratin flour is also the largest. The deviation value for variant V4 is 1.53.

### 3.2. Thermal Resistance Model

If the laminate is made of several layers with different thermal conductivity coefficients, the transfer coefficient is calculated by determining the total thermal resistance:(6)$$R=\sum_{i=1}^{n}R_{i}=\sum_{i=1}^{n}\frac{d_{i}}{\lambda i}$$
where *n*—number of layers of laminate; *i*—layer number.

The reciprocal/inverse of this resistance is the heat transfer coefficient we are looking for. This formula applies to homogeneous materials. There are many studies showing the functional relationship between thermal conductivity and relative humidity of various materials [23]. However, in the case of biolaminates, this issue should be approached more analytically. The best thermal resistance model in this case would be the modified Ju Wie model [24].
(7)$$R={\frac{D}{\lambda }}_{air}\left(\frac{\lambda_{wet\;laminate}-a(\lambda_{wet\;laminate}-\lambda_{air})}{\lambda_{wet\;laminate}-(\lambda_{wet\;laminate}-\lambda_{air})(\alpha -F_{wet\;laminate}}\right)$$
where

*R_laminate_*—laminate thermal resistance m^2^ KW^−1^;

*D*—laminate thickness, m;

*λ_air_*—air thermal conductivity 0.0226 W·m^−1^ K^−1^;

*a*—laminate structural parameters; *a* = *D_compressed_*/*D*;

*D_compressed_*—thickness measured at 15 kPa pressure, m;

*D*—thickness measured at 2 kPa pressure, m;

*F*—laminates filling coefficient, %.

The laminates filling coefficient is calculated using the following formula:(8)$$F=\frac{VD}{Fiber\;density},\%$$
(9)$$\mathrm{VD}=\frac{Laminates\;Areal\;Density}{Fabric\;Thickness},[{\mathrm{kgm}}^{-3}]$$

VD—volumetric density, kgm^−3^;

Areal density, gm^−2^.

### 3.3. Discussions

This work involved measuring the thermal insulation properties of laminates in accordance with standards: thermal resistance and thermal conductivity. These coefficients were analyzed for two variants of ambient relative humidity and two different variants of mass moisture transport. The smaller the difference between the thermal resistance of the tested laminate measured for individual variants of relative humidity and the resistance when a mass of moisture was added to its bottom layer, the greater the physiological comfort of the tested laminate would be. Comparing the obtained results (Figure 5), it could be concluded that the higher the content of keratin flour in the biolaminate, the smaller the difference in thermal resistance for both variants of relative humidity and the supplied mass of moisture. The smallest differences at 0 mL of supplied moisture occurred for the variant with 20 g/m^2^ of keratin flour, and at 1.5 mL of supplied moisture, the smallest difference was for the variant of 60 g/m^2^. The values of these coefficients were adopted as criteria for the physiological comfort of the laminate in the conditions covered by the tests.

The Ju Wie thermal resistance model was proposed as a tool that enables the easy design of such laminates in terms of thermal insulation properties. According to the model, the coefficient of determination R^2^ allows us to assess the quality of fit of the tested model to the data. In this result presented in Figure 6, the coefficient R^2^ is at an acceptable level and ranges from 0.89 to 0.97.

## 4. Conclusions

The presented research shows that it is technically possible to produce biomaterials from waste—keratin flour. The aim of the research was to determine the effect/influence of the relative humidity of the environment and the mass of moisture supplied to the bottom layer of the biolaminate on the values of its thermal resistance and thermal conductivity.

Managing keratin waste is a multifaceted challenge that requires a combination of innovative technologies, sustainable practices, and regulatory compliance to ensure environmental protection and resource recovery. It is important to note that the reuse of keratin powder may vary depending on the specific application and the quality of the keratin.

Covering the nonwoven fabric with keratin flour significantly improved the physiological comfort parameters of the tested laminate at variable values of relative ambient humidity and at the various states of moisture mass supplied to its bottom layer. This was manifested by an increase in the value of the thermal resistance coefficient. It was noticed that for samples of similar thickness, the highest thermal resistance was observed for the sample with a keratin flour layer of 60 g/m^2^. From among the analyzed laminates, the best thermal insulation properties, tested at variable values of the relative humidity of the external environment and the mass of moisture supplied to the bottom layer, were characterized by the laminate that consisted of a nonwoven fabric with a water-repellent finish covered with keratin flour with a basis weight of 60 g/m^2^. After exceeding the density of nonwoven fabric filling with keratin flour, equal to 20 g/m^2^, the resistance to delamination of the laminate adhesive joint decreased. The laboratory tests showed that the optimized density of nonwoven fabric filling with keratin flour, due to the laminate’s resistance to delamination and its thermal insulation properties, was in the range of 20–40 g/m^2^. To further improve the physiological comfort of the laminate, a change in technology would be necessary:Hydrophilization of the top layer of the laminate;Gluing the creatine grain to the laminate with the nonwoven fabric.

Thermal conductivity λ is a physical property of each material, and it is not a constant value. It depends on factors such as structure, pressure, temperature, density, and humidity. The smaller the λ, the better the material is as an insulator. In turn, the higher the value of the thermal resistance coefficient, the better the insulating properties of the material. A comparison of the thermal insulation properties of the tested composites with other types of biomaterials was presented. The results showed good thermal insulation properties of the created biolaminate made of keratin flour, e.g., thermal conductivity, and λ of wool is 0.0385–0.040 W/mK. The wood fiber insulation board has a thermal conductivity of 0.035 W/mK; hemp products of 0.05 W/mK; mussel shells of 0.12 W/mK, and a composite made of wine waste of 0.35 W/mK [25].

This type of approach to textile functionalization is a part of the upcycling and circular economy process. While the production of keratin and the use of biolaminates have environmental impacts, there are promising strategies to mitigate these effects. Biotechnological advances, renewable energy, and regulatory measures can improve the sustainability of keratin production.

Processing wood waste into biopolymers and biocomposites provides a sustainable alternative to traditional plastics, further supporting the transition to a circular economy, although further research and comparisons with conventional materials and assessment of environmental benefits are necessary.

## Figures and Tables

**Figure 1 materials-17-04081-f001:**
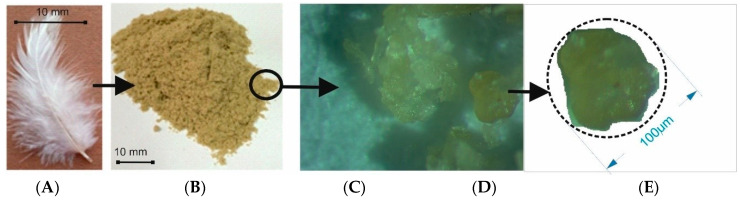
Keratin flour as a component of textile laminates. (**A**) Close-up image of a chicken feather and (**B**) keratin flour (**C**) single grain—magnification ×40; (**D**) substitute diameter of keratin flour grain; (**E**) an example photo of the biolaminate surface with keratin grains, taken with the use of a scanning microscope; lens: MPLAPONLEXT100, zoom: 5×, image size: 220 × 210 µm.

**Figure 2 materials-17-04081-f002:**
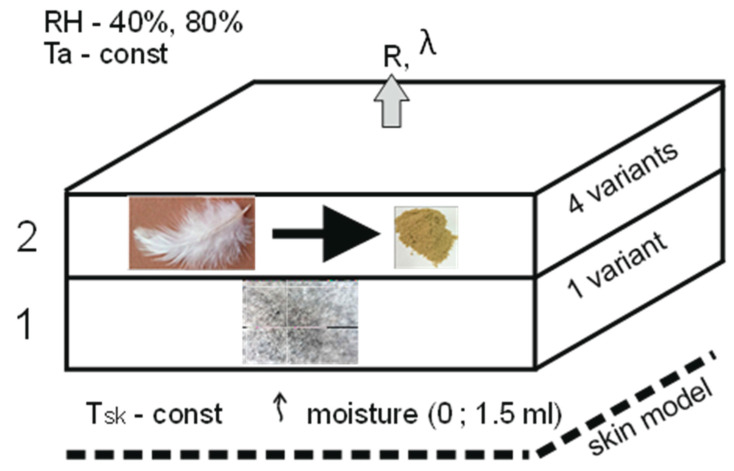
Qualitative measurement model, where 1, 2—laminate layers; Ta—ambient temperature; RH—relative humidity; Tsk—constant skin temperature; moisture—amount of sweat: 0 mL ÷ 1.5 mL; R—thermal resistance; λ—thermal conductivity coefficient.

**Figure 3 materials-17-04081-f003:**
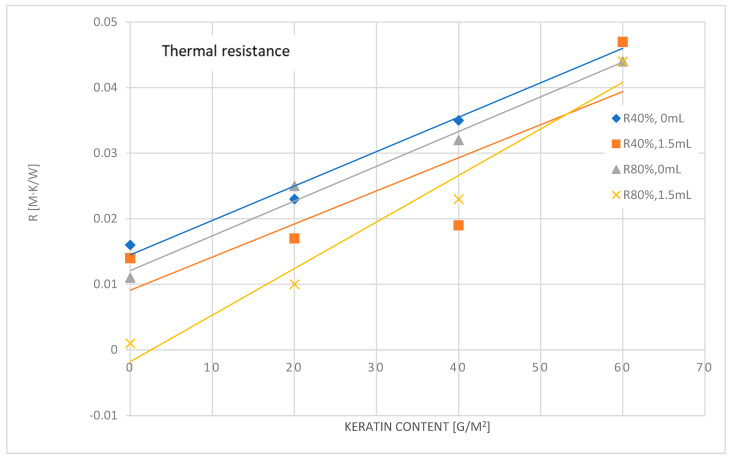
Characteristics of the thermal resistance coefficient as a function of the amount of keratin content in biolaminate.

**Figure 4 materials-17-04081-f004:**
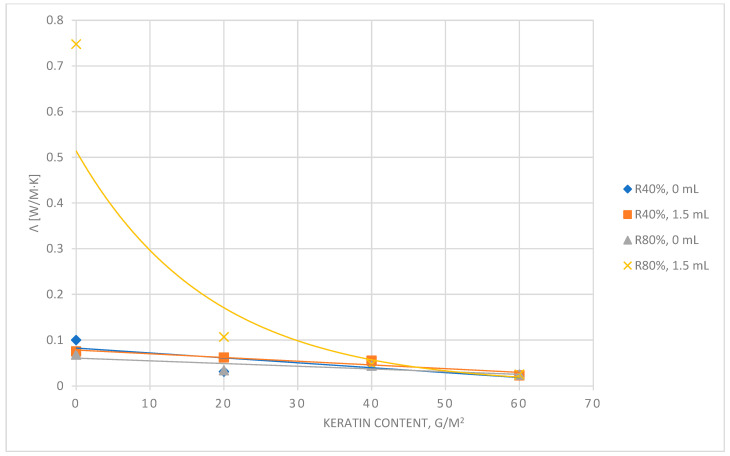
Characteristics of the thermal conductivity coefficient as a function of the amount of creatine keratin in biolaminate.

**Figure 5 materials-17-04081-f005:**
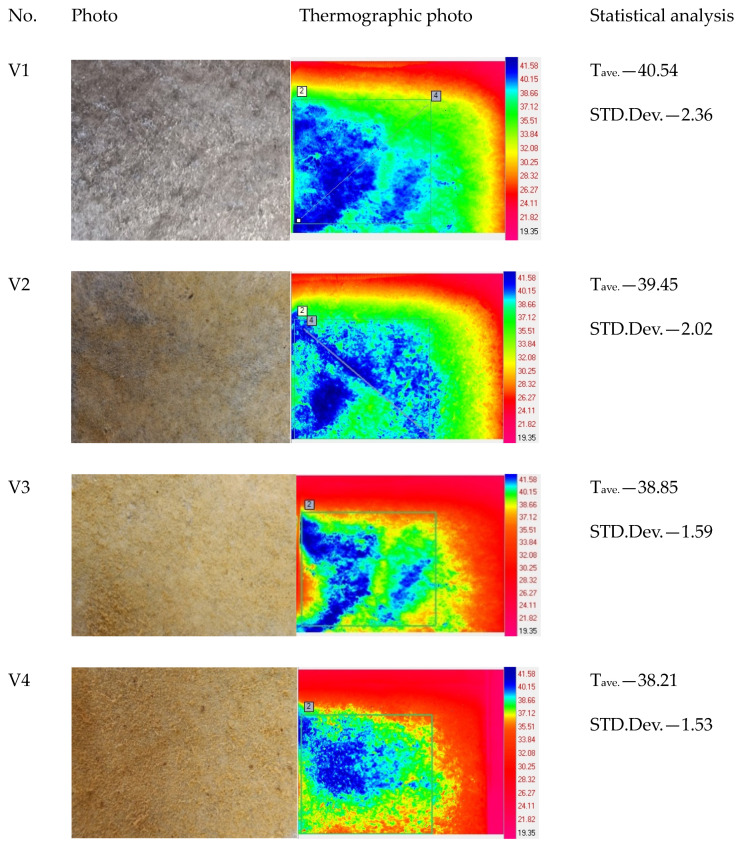
Thermographic photo of individual samples and temperature distribution results; T_ave_.—average temperature of laminates; cold color palette with temperature scale. Number 2 and 4 analyzed areas.

**Figure 6 materials-17-04081-f006:**
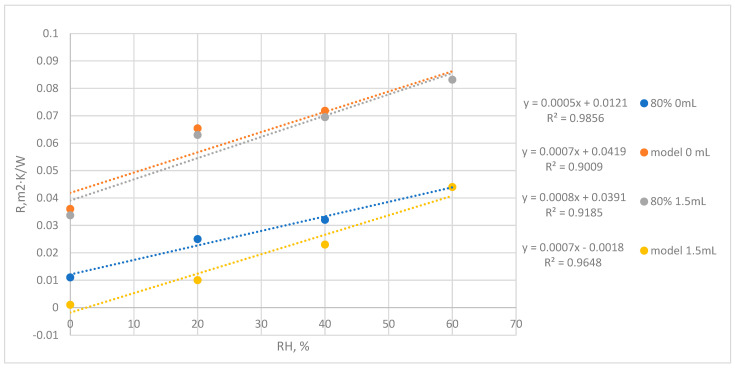
Theoretical thermal resistance compared to experimental at different humidity levels (RH 80% 0–1.5 mL). Dark blue and green curve: experimental results; orange and light blue curve: model results.

**Table 1 materials-17-04081-t001:** (**A**) Characteristics of the basic properties of the 1st layer of the biolaminate–polyester nonwoven fabric, together with the test methodology. (**B**) Characteristics of the basic properties of the laminate with keratin flour.

(**A**)				
**The Tested Parameters of the Polyester Nonwoven Fabric + Keratin Flour**		**Standards**	**Parameters**	**Units**
Average area weight		PN-EN ISO 9073-1:2023-11 [17]	113 ± 16	g/m^2^
Thickness		PN-EN ISO 9073-2:2002 [18]	0.71 ± 0.07	mm
Liquid absorption		PN-EN ISO 9073-6 [19]	547 ± 44	g/m^2^
Water sorption		PN-EN ISO 9073-6:2005 [19]	1.49	μL/cm^2^
Max sorption speed		Research Procedure No. 14/1:2003 edition I	1.11	μL/cm^2^ s
Average speed sorption		Research Procedure No. 14/1:2003 edition I	0.315	μL/cm^2^ s
Total sorption time		Research Procedure No. 14/1:2003 edition I	45.5	s
Bacterial resistance		Hygienic certificate no385/322/413/2015	positive	
Adhesion test		ISO 2409; Adhesion test tape method [20]	Positive evaluation for all sample variants	
(**B**)				
**No.**	**The Share of Keratin Flour in a Second Layer of the Laminate g/m^2^**	**Laminate Thickness d, 10^−3^ m**	**Laminate Surface Mass Mp, g/m^2^**	
V1	0	0.71	113	
V2	20	0.98	132	
V3	40	1.04	154	
V4	60	1.15	179	

**Table 2 materials-17-04081-t002:** (A) Results of thermal conductivity and thermal resistance measurements of the tested laminates with 0% of delivered sweat. (B) Results of thermal conductivity and thermal resistance measurement of the tested laminates with 1.5 mL delivered sweat.

Lp	g/m^2^	Thermal Resistance R [m^2^ K/W] Measured for Two Variants of Relative Humidity RH		Thermal Conductivity Coefficient λ [W/mK] Measured for Two Variants of Relative Humidity RH	
A		80%	40%	80%	40%
V1	0	0.011	0.016	0.068	0.100
V2	20	0.025	0.023	0.034	0.031
V3	40	0.032	0.035	0.044	0.047
V4	60	0.044	0.047	0.025	0.023
B					
V1	0	0.001	0.014	0.748	0.075
V2	20	0.010	0.017	0.107	0.062
V3	40	0.023	0.019	0.047	0.055
V4	60	0.044	0.047	0.025	0.023

## Data Availability

The data presented in this study are available upon request to the corresponding author due to ongoing research.

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
