# Peer review of "Biolaminates as an Example of Upcycling Product with Keratin Flour—Research and Thermal Properties Modeling"

_materials, 2024, doi:10.3390/ma17164081_

Round 1
Reviewer 1 Report
Comments and Suggestions for Authors
The manuscript reports a material combining keratin upcycled from poultry waste for producing textile laminates. The work could be of interest for publication but requires extensive changes before it is ready for publication. Please find enclosed my point-by-point arguments here:
1) The arguments in the text must be provided with representative numbers to illustrate the problem more scientifically. For instance, what is the exact amount of poultry waste produced globally? Which pollutants are released into the environment?
2) The novelty of the work needs to be clearer. The use of keratin as a thermal insulator has been previously reported in the literature. Also, even within the experimental, several references are provided for preparing the materials based on previous works. Please clarify this.
3) What is the elemental composition of the flour used for the materials?
4) Figure 1: Are these the results of this work? If so, why are they placed in the experimental section? Scale bars are also missing in A, B, and E.
5) The work does not include statistical analysis of the data, and no standard deviations of the data are presented. Please revise this.
6) What skin model was used?
7) How were the samples isolated to ensure minimal loss of heat during the experiments? The experimental section needs to include much more details about the experimental preparation.
8) Why were microstructural analysis images not included in the work? This is very important in biocomposite preparation, especially when assessing physicochemical properties, as the presence of voids, for instance, can impact the discussions provided.
9) How does this work contribute to a circular economy? Which environmental assessment indicator was used to evaluate its impact?
Comments on the Quality of English LanguageNeeds English proof from a professional service.
Author Response
Dear reviewer,
Thank you for your time and valuable comments. I would like to refer to some of them:
1) The arguments in the text must be provided with representative numbers to illustrate the problem more scientifically. For instance, what is the exact amount of poultry waste produced globally? Which pollutants are released into the environment?
It was added in the main text:
In the article [Zhang L. 2023] describes the threats resulting from the constant increase in poultry production. Feather waste is byproduct that is massively generated by the poultry industry globally Bird feathers are mainly composed of keratin, which is an insoluble, fibrous, recalcitrant protein, and it accounts for 85% to 99% of the total dried feather weight However, feather waste is also a serious concern that may lead to environmental pollution because a lot of pathogens and microorganisms such as Salmonella and Vibrio can be found in poultry feathers. Various pollutants, such as ammonia, nitrous oxide, and hydrogen sulfide, can also be emitted from feather waste, which has brought about threats and risks to both human health and environmental safety [Zhang L. 2023, Karuppannan S.K,, 2020].
And citations
Zhang, Long, Jingzheng Ren, and Wuliyasu Bai. 2023. "A Review of Poultry Waste-to-Wealth: Technological Progress, Modeling and Simulation Studies, and Economic- Environmental and Social Sustainability" Sustainability 15, no. 7: 5620. https://doi.org/10.3390/su15075620
Karuppannan, S.K.; Dowlath, M.J.H.; Raiyaan, G.I.D.; Rajadesingu, S.; Arunachalam, K.D. Application of poultry industrywaste in producing value-addedproducts—A review. In Concepts of Advanced Zero Waste Tools: Present and Emerging Waste Management Practices; Hussain, C.M., Ed.; Elsevier Inc.: Amsterdam, The Netherlands, 2020; pp. 91–121.
- The novelty of the work needs to be clearer. The use of keratin as a thermal insulator has been previously reported in the literature. Also, even within the experimental, several references are provided for preparing the materials based on previous works. Please clarify this.
The article presents a new type of material with increased thermal insulation properties. This material is a laminate in which one of the layers is keratin flour. This type of material combination, typical for laminate technology, is known, but there are no reports of the use of keratin flour as one of the layers.
In the "Materials and Methods" section I quote reports on creating materials from flour and show the differences. Quotes:
Similar work has been conducted by other researchers in that area [Verma Det al. 2013, Wrześniewska-Tosik K. , 2020]. However in these cited work, most of the flour has been added to the thermoplastic materials by the authors, creating a kind of polymeric matrix of the biocomposite. Another works have been also carried out on the inclusions of keratin flour directly into the structures of nonwovens(in thermoplastic fibers structure), but have not involved the modification of the nonwovens themselves and have not included the study of the thermo-properties of the material (ISO 11092:2014). [Shavandi A. et al. 2019]. In presented article, laminate consists of a hydrophilic non-woven fabric on the side for the back sheet. On the other side, it was covered with keratin flour. Various variants of the mass fractions of meal charges in the package were tested.
- What is the elemental composition of the flour used for the materials?
Detailed characteristics of the raw material (keratin flour) are provided by the manufacturer. [www.kemos.com.pl/] In response to the reviewer, I am sending an extensive description of the flour's characteristics. Selected information in the text of the article:
Feather hydrolyzate is made in the process of feathers thermohydrolysis under the pressure of 5 bars in order to cleave keratin bonds. Technology applied in this process permits to obtain the product of high protein digestibility. Share of blood and feather is analogical to share in living bird. Powder contains 100% poultry substance. Stock for production comes down from healthy animals (mixed stock – poultry, turkey, goose), which meat was accepted by Veterinary Inspection as good to consume for people. Powder is stabilized with positive antioxidant BHA, BHT. There is no etoksyquina (E324) and any other antioxidant. [ www.kemos.com.pl/]
- Figure 1: Are these the results of this work? If so, why are they placed in the experimental section? Scale bars are also missing in A, B, and E.
Figure 1 contains the author's own photos. Their purpose is only to show the raw material and final product. In the author's opinion these photos shouldn’t be included in the research part because they do not contribute anything to the experimental plan. However, the scale in the photos has been added.
- The work does not include statistical analysis of the data, and no standard deviations of the data are presented. Please revise this.
Thank you for this tip. in fact, statistical analysis was given in the article only for thermal imaging measurements. The rest of the results were also statistically analyzed.
- What skin model was used?
This is measuring equipment, standardized : Skin Model, PN-EN 31092:1998/A1:2013-03E ( it was added)
- How were the samples isolated to ensure minimal loss of heat during the experiments? The experimental section needs to include much more details about the experimental preparation.
It was added: Thermal tests were performed in accordance with the standard: PN-EN 31092:1998/A1:2013-03E. Textiles - Determination of physiological properties - Measurement of thermal resistance and water vapor resistance under steady-state conditions. (thermally insulated sweating plate method).
Detailed experiment was described in section 2.1. Measurement Methodology.
8) Why were microstructural analysis images not included in the work? This is very important in biocomposite preparation, especially when assessing physicochemical properties, as the presence of voids, for instance, can impact the discussions provided.
Thermographic analysis was carried out in this work. The work focuses on thermal insulation properties and on the proposed model of these properties. Microstructural analysis will be presented in the next articles.
- ) How does this work contribute to a circular economy? Which environmental assessment indicator was used to evaluate its impact?
it was added: . While the production of creatine and the use of biolaminates have environmental implications, there are promising strategies to mitigate these impacts. Biotechnological advancements, renewable energy, and regulatory measures can improve the sustainability of creatine production.
The valorization of wood waste into biopolymers and biocomposites represents a sustainable alternative to traditional plastics, further supporting the transition to a circular economy. Although it is need of further research and comparisons with conventional material and the assessment of environmental impact benefits.
Best regards,
Author
Reviewer 2 Report
Comments and Suggestions for Authors
The paper reviewed deals with the management of keratin waste, specifically in the form of flour, originating mainly from poultry processing industries. This waste is exploited as an upcycling material for the production of biolaminates with thermal insulation properties. The study focuses on the applicability of keratin flour in textile laminates with a view to the improvement of thermal comfort and thermal resistance which is relevant to both industrial applications and consumption of resources.
Strengths:
-The study addresses an innovative application of an industrial waste product, contributing to the CE and environmental sustainability.
-The methodology employed is rigorous and well detailed thus providing a solid basis for the replication of results.
-The analysis includes a variety of experimental conditions such as different levels of humidity and amounts of sweat which enhance the understanding of properties in biolaminates.
Weaknesses:
-The use of keratin flour may be limited by availability and also the process of obtaining it which could be an obstacle to large-scale production.
-The article could benefit from a direct comparison between keratin biolaminates and other conventional insulating materials to clearly highlight the advantages of the new material.
-Despite the focus is on the reuse of waste, no quantitative data is presented on the impact on the environment of the production and use of biolaminates that could strengthen the argument regarding environmental benefits.
-There is no research question and should be added.
-There is no particularly well-founded literature review and should be considered.
The article presented shows an advance in the use of keratin waste for the production of biolaminates with good thermal insulation, representing a helpfull contribution to sustainable production practices.
Although there are limitations (i.e., the need of further comparisons with conventional material and the assessment of environmental impact) and important issues to be corrected - such as the absence of any research question and a more elaborated state of the art.
Author Response
Dear reviewer,
Thank you for your time and valuable comments. I would like to refer to some of them:
-The use of keratin flour may be limited by availability and also the process of obtaining it which could be an obstacle to large-scale production.
It was added:
Biolaminates are emerging as a revolutionary material in various industries due to their unique properties and environmental benefits.
One of the most significant benefits of biolaminates is their sustainability. They are made from renewable biological resources, which reduces reliance on non-renewable fossil-based materials
. By replacing conventional plastics with more sustainable bioplastics, biolaminates contribute to a greener manufacturing process and product lifecycle
-The article could benefit from a direct comparison between keratin biolaminates and other conventional insulating materials to clearly highlight the advantages of the new material.
It was added: A comparison of the thermal insulation properties of the tested composites with other types of biomaterials was demonstrated. This showed good thermal insulation properties of created biolaminate with keratine flour. E.g. thermal conductivity for wool is 0.0385–0.040 W/mK. For wood fiber insulation board has a thermal conductivity of 0.035 W/mK; Hemp products showed 0.05 W/mK; mussel shell (TC of 0.12 W/mK), or wine waste composite (TC of 0.35 W/mK) [Cosentino, L 2023].
Despite the focus is on the reuse of waste, no quantitative data is presented on the impact on the environment of the production and use of biolaminates that could strengthen the argument regarding environmental benefits.
It was added (page 2) ).
In the article [Zhang L. 2023] describes the threats resulting from the constant increase in poultry production. Feather waste is byproduct that is massively generated by the poultry industry globally Bird feathers are mainly composed of keratin, which is an insoluble, fibrous, recalcitrant protein, and it accounts for 85% to 99% of the total dried feather weight. However, feather waste is also a serious concern that may lead to environmental pollution because a lot of pathogens and microorganisms such as Salmonella and Vibrio can be found in poultry feathers. Various pollutants, such as ammonia, nitrous oxide, and hydrogen sulfide, can also be emitted from feather waste, which has brought about threats and risks to both human health and environmental safety [Zhang L. et al. 2023, Karuppannan S.K. et al., 2020].
(page 20)
While the production of creatine and the use of biolaminates have environmental implications, there are promising strategies to mitigate these impacts. Biotechnological advancements, renewable energy, and regulatory measures can improve the sustainability of creatine production.
The valorization of wood waste into biopolymers and biocomposites represents a sustainable alternative to traditional plastics, further supporting the transition to a circular economy. Although it is need of further research and comparisons with conventional material and the assessment of environmental impact benefits.
-There is no research question and should be added.
It was added: The aim of the research was to determine the effect of the relative humidity of the environment and the mass of moisture supplied to the bottom layer of the biolaminate on its values ​​of thermal resistance and thermal conductivity.
-There is no particularly well-founded literature review and should be considered.
It was added new literature:
Cetiner, I.; Shea, A.D. Wood Waste as an Alternative Thermal Insulation for Buildings. Energy Build. 2018, 168, 374–384
Cosentino, L.; Fernandes, J.; Mateus, R. A Review of Natural Bio-Based Insulation Materials. Energies 2023, 16, 4676. https://doi.org/10.3390/en16124676
Zhang, Long, Jingzheng Ren, and Wuliyasu Bai. 2023. "A Review of Poultry Waste-to-Wealth: Technological Progress, Modeling and Simulation Studies, and Economic- Environmental and Social Sustainability" Sustainability 15, no. 7: 5620. https://doi.org/10.3390/su15075620
Karuppannan, S.K.; Dowlath, M.J.H.; Raiyaan, G.I.D.; Rajadesingu, S.; Arunachalam, K.D. Application of poultry industrywaste in producing value-addedproducts—A review. In Concepts of Advanced Zero Waste Tools: Present and Emerging Waste Management Practices; Hussain, C.M., Ed.; Elsevier Inc.: Amsterdam, The Netherlands, 2020; pp. 91–121.
Comparisons with conventional material :
It was added:
A comparison of the thermal insulation properties of the tested composites with other types of biomaterials was demonstrated. This showed good thermal insulation properties of created biolaminate with keratine flour. E.g. thermal conductivity for wool is 0.0385–0.040 W/mK. For wood fiber insulation board has a thermal conductivity of 0.035 W/mK; Hemp products showed 0.05 W/mK; mussel shell (TC of 0.12 W/mK), or wine waste composite (TC of 0.35 W/mK) [Cosentino, L 2023].
Best regards,
Author
Reviewer 3 Report
Comments and Suggestions for Authors
After review the manuscript, this need to be improved significantly due it seems to be a technical report more than a scientific manuscript.
The title indicate the use of keratin flour for biolaminates, but there is a lack of experimental design or information about how the biolaminates were ellaborated, for instance keratin flour content in laminates, also it is not indicated the main material for laminates (only indicate that thermoplastic materials are used, but which kind of thermoplastic material?).
Methodology needs to be improved due it is not clear about how the materials were obtained and how the test were carried out. For instance, How would be evaluated the criterion for choosing the optimal laminate structure?? The mentionated criterion are theoretical and it is not indicated a experimental test for evaluation. Also, title of subsection 2.1 must corresponds to technique explained.
only thermal resistance coefficient and thermal conductivity coefficient were al the results in this work?
In page 7 the table has not caption.
Figures 3 and 4 caption indicate that thermal resistance coeficient is reported in functino of creatine content. Also in x figure xis indicate ceratine, and caption indicate creatine.
Data reported as table 3 i recommend to change to figures.
Also photos reported in it, how those photos were obtained and which is the magnification??
I recommend to delete the caption inserted upper of the figure 5, also please indicate which curve corresponds to theoretical and which to experimental value due it is not clear enough.
Please follow the instructions for authors of journal for cite references in main text and to report references.
In general the work has a lack of experimental results and discussion.
Author Response
Dear Reviewer,
Thank you for your time and valuable comments. I would like to refer to some of them:
After review the manuscript, this need to be improved significantly due it seems to be a technical report more than a scientific manuscript.
It was added research aim:
The aim of the research was to determine the effect of the relative humidity of the environment and the mass of moisture supplied to the bottom layer of the biolaminate on its values ​​of thermal resistance and thermal conductivity.
The title indicate the use of keratin flour for biolaminates, but there is a lack of experimental design or information about how the biolaminates were ellaborated, for instance keratin flour content in laminates, also it is not indicated the main material for laminates (only indicate that thermoplastic materials are used, but which kind of thermoplastic material?).
It was added:
Laminate consists of a hydrophilic non-woven fabric on the side for the back sheet. On the other side, it was covered with keratin flour. Various variants of the mass fractions of meal charges in the package were tested.
Prameters of bioaminate was described in Table 1: Table 1. A) Characteristics of the basic properties of the 1st layer of biolaminate-polyester non-woven fabric, together with the research methodology. B) Characteristics of the basic properties of the laminate with keratin flour
Description of keratin flour was added:
Feather hydrolyzate is made in the process of feathers thermohydrolysis under the pressure of 5 bars in order to cleave keratin bonds. Technology applied in this process permits to obtain the product of high protein digestibility. Share of blood and feather is analogical to share in living bird. Powder contains 100% poultry substance. Stock for production comes down from healthy animals (mixed stock – poultry, turkey, goose), which meat was accepted by Veterinary Inspection as good to consume for people. Powder is stabilized with positive antioxidant BHA, BHT. There is no etoksyquina (E324) and any other antioxidant. [ www.kemos.com.pl/].
Methodology needs to be improved due it is not clear about how the materials were obtained and how the test were carried out. For instance, How would be evaluated the criterion for choosing the optimal laminate structure?? The mentionated criterion are theoretical and it is not indicated a experimental test for evaluation. Also, title of subsection 2.1 must corresponds to technique explained.
According to the questions:
- … how the materials were obtained and how the test were carried out
A satisfactory level of hydrophobization of the polyester non-woven fabric was obtained by modifying its surface with hydrophobic agents. It was padded at 150°C in an aqueous dispersion of dodecyldimethylamine. The concentration of 40 ml per 1 liter of water was assumed to be favorable, based on the conducted research. The commercially available Kemiline Guard YSA from Kemitekst (Istanbul, Turkey) was used as a hydrophilizing formulation. The keratin flour was bonded to the surface of textiles with the use of a TensorGrip L14 contact adhesive spray, which has negligible heat resistance. Taking into account the assumptions of the laminate structure and the above-described materials, the subject of the research were laminates made of a polyester non-woven top layer with a hydrophobic coating and, in the bottom layer, optionally with keratin powder with four variants of the backfill density.
..how the test were carried out:
-Table 1A;
-it was added: Thermal tests were performed in accordance with the standard: PN-EN 31092:1998/A1:2013-03E. Textiles - Determination of physiological properties - Measurement of thermal resistance and water vapor resistance under steady-state conditions. (thermally insulated sweating plate method);
- Paragraph 2.2. e.g. page 11-12
How would be evaluated the criterion for choosing the optimal laminate structure?
The selection criterion is thermal insulation properties, the research was based on a qualitative model that assumes 4 test variants, 2 variable values ​​and process constants. Figure 2 presents Qualitative measurement model.
only thermal resistance coefficient and thermal conductivity coefficient were al the results in this work?
The aim of the test was to determine the thermal insulation properties of laminates made of keratin flour and thermal properties modeling.
only thermal resistance coefficient and thermal conductivity coefficient were al the results in this work?
According to standard PN-EN 31092:1998/A1:2013-03E these parameters are important. Thermal tests were performed in accordance with the standard: PN-EN 31092:1998/A1:2013-03E. Textiles - Determination of physiological properties - Measurement of thermal resistance and water vapor resistance under steady-state conditions. (thermally insulated sweating plate method).
In page 7 the table has not caption. The caption was changed.
Table 1. A) Characteristics of the basic properties of the 1st layer of bio laminate-polyester non-woven fabric, together with the research methodology. B) Characteristics of the basic properties of the laminate with keratin flour
Figures 3 and 4 caption indicate that thermal resistance coeficient is reported in functino of creatine content. Also in x figure xis indicate ceratine, and caption indicate creatine. Data reported as table 3 i recommend to change to figures.
It was changed. Figure 3. Characteristics of the thermal resistance coefficient as a function of the amount of creatine content in biolaminate
Also photos reported in it, how those photos were obtained and which is the magnification??
It was added: :… It was used Flir SC 5000, 320x256 pixel resolution, 0.02K sensivity, 150 Hz acquision frequency.
I recommend to delete the caption inserted upper of the figure 5, also please indicate which curve corresponds to theoretical and which to experimental value due it is not clear enough.
It was changed:
Figure 6. Theoretical thermal resistance vs experimental at different moisture levels(RH80% 0ml - 1,5ml): dark blue and green curve – experimental results; orange and light blue curve – model results.
The coefficient of determination R2 allows you to assess the quality of fit of the tested model to the data. In this result presented in fig.6. coefficient R2 is at an acceptable level and varies in the range 0.89 – 0.97.
In general the work has a lack of experimental results and discussion.
It was added comparison I Discussions section:
… The thermal conductivity λ is a physical property of each material and it is not a constant value. It depends on factors such as: structure, pressure, temperature, density and humidity. The smaller λ, the material is a better insulator. In turn, the higher value of the thermal resistance coefficient, the better the material's insulating properties. A comparison of the thermal insulation properties of the tested composites with other types of biomaterials was demonstrated. This showed good thermal insulation properties of created biolaminate with keratine flour. E.g. thermal conductivity for wool is 0.0385–0.040 W/mK. For wood fiber insulation board has a thermal conductivity of 0.035 W/mK; Hemp products showed 0.05 W/mK; mussel shell (TC of 0.12 W/mK), or wine waste composite (TC of 0.35 W/mK) [Cosentino, L 2023].
Round 2
Reviewer 1 Report
Comments and Suggestions for Authors
No more comments to add.
Comments on the Quality of English LanguageAn English-proof service should revise the text.
Author Response
Dear Reviewer,
Thank you for your time and valuable comments. I would like to refer to some of them:
1) The arguments in the text must be provided with representative numbers to illustrate the problem more scientifically. For instance, what is the exact amount of poultry waste produced globally? Which pollutants are released into the environment?
It was added in the main text:
In the article [Zhang L. 2023] describes the threats resulting from the constant increase in poultry production. Feather waste is byproduct that is massively generated by the poultry industry globally Bird feathers are mainly composed of keratin, which is an insoluble, fibrous, recalcitrant protein, and it accounts for 85% to 99% of the total dried feather weight However, feather waste is also a serious concern that may lead to environmental pollution because a lot of pathogens and microorganisms such as Salmonella and Vibrio can be found in poultry feathers. Various pollutants, such as ammonia, nitrous oxide, and hydrogen sulfide, can also be emitted from feather waste, which has brought about threats and risks to both human health and environmental safety [Zhang L. 2023, Karuppannan S.K,, 2020].
And citations
Zhang, Long, Jingzheng Ren, and Wuliyasu Bai. 2023. "A Review of Poultry Waste-to-Wealth: Technological Progress, Modeling and Simulation Studies, and Economic- Environmental and Social Sustainability" Sustainability 15, no. 7: 5620. https://doi.org/10.3390/su15075620
Karuppannan, S.K.; Dowlath, M.J.H.; Raiyaan, G.I.D.; Rajadesingu, S.; Arunachalam, K.D. Application of poultry industrywaste in producing value-addedproducts—A review. In Concepts of Advanced Zero Waste Tools: Present and Emerging Waste Management Practices; Hussain, C.M., Ed.; Elsevier Inc.: Amsterdam, The Netherlands, 2020; pp. 91–121.
- The novelty of the work needs to be clearer. The use of keratin as a thermal insulator has been previously reported in the literature. Also, even within the experimental, several references are provided for preparing the materials based on previous works. Please clarify this.
The article presents a new type of material with increased thermal insulation properties. This material is a laminate in which one of the layers is keratin flour. This type of material combination, typical for laminate technology, is known, but there are no reports of the use of keratin flour as one of the layers.
In the "Materials and Methods" section I quote reports on creating materials from flour and show the differences. Quotes:
Similar work has been conducted by other researchers in that area [Verma Det al. 2013, Wrześniewska-Tosik K. , 2020]. However in these cited work, most of the flour has been added to the thermoplastic materials by the authors, creating a kind of polymeric matrix of the biocomposite. Another works have been also carried out on the inclusions of keratin flour directly into the structures of nonwovens(in thermoplastic fibers structure), but have not involved the modification of the nonwovens themselves and have not included the study of the thermo-properties of the material (ISO 11092:2014). [Shavandi A. et al. 2019]. In presented article, laminate consists of a hydrophilic non-woven fabric on the side for the back sheet. On the other side, it was covered with keratin flour. Various variants of the mass fractions of meal charges in the package were tested.
- What is the elemental composition of the flour used for the materials?
Detailed characteristics of the raw material (keratin flour) are provided by the manufacturer. [www.kemos.com.pl/] In response to the reviewer, I am sending an extensive description of the flour's characteristics. Selected information in the text of the article:
Feather hydrolyzate is made in the process of feathers thermohydrolysis under the pressure of 5 bars in order to cleave keratin bonds. Technology applied in this process permits to obtain the product of high protein digestibility. Share of blood and feather is analogical to share in living bird. Powder contains 100% poultry substance. Stock for production comes down from healthy animals (mixed stock – poultry, turkey, goose), which meat was accepted by Veterinary Inspection as good to consume for people. Powder is stabilized with positive antioxidant BHA, BHT. There is no etoksyquina (E324) and any other antioxidant. [ www.kemos.com.pl/]
- Figure 1: Are these the results of this work? If so, why are they placed in the experimental section? Scale bars are also missing in A, B, and E.
Figure 1 contains the author's own photos. Their purpose is only to show the raw material and final product. In the author's opinion these photos shouldn’t be included in the research part because they do not contribute anything to the experimental plan. However, the scale in the photos has been added.
- The work does not include statistical analysis of the data, and no standard deviations of the data are presented. Please revise this.
Thank you for this tip. in fact, statistical analysis was given in the article only for thermal imaging measurements. The rest of the results were also statistically analyzed.
- What skin model was used?
This is measuring equipment, standardized : Skin Model, PN-EN 31092:1998/A1:2013-03E ( it was added)
- How were the samples isolated to ensure minimal loss of heat during the experiments? The experimental section needs to include much more details about the experimental preparation.
It was added: Thermal tests were performed in accordance with the standard: PN-EN 31092:1998/A1:2013-03E. Textiles - Determination of physiological properties - Measurement of thermal resistance and water vapor resistance under steady-state conditions. (thermally insulated sweating plate method).
Detailed experiment was described in section 2.1. Measurement Methodology.
8) Why were microstructural analysis images not included in the work? This is very important in biocomposite preparation, especially when assessing physicochemical properties, as the presence of voids, for instance, can impact the discussions provided.
Thermographic analysis was carried out in this work. The work focuses on thermal insulation properties and on the proposed model of these properties. Microstructural analysis will be presented in the next articles.
- ) How does this work contribute to a circular economy? Which environmental assessment indicator was used to evaluate its impact?
it was added: . While the production of creatine and the use of biolaminates have environmental implications, there are promising strategies to mitigate these impacts. Biotechnological advancements, renewable energy, and regulatory measures can improve the sustainability of creatine production.
The valorization of wood waste into biopolymers and biocomposites represents a sustainable alternative to traditional plastics, further supporting the transition to a circular economy. Although it is need of further research and comparisons with conventional material and the assessment of environmental impact benefits.
Best regards,
Author
Reviewer 2 Report
Comments and Suggestions for Authors
Paper is improved and now ok.
Author Response
Dear Reviewer,
Thank you for your time and valuable comments. I would like to refer to some of them:
-The use of keratin flour may be limited by availability and also the process of obtaining it which could be an obstacle to large-scale production.
It was added:
Biolaminates are emerging as a revolutionary material in various industries due to their unique properties and environmental benefits.
One of the most significant benefits of biolaminates is their sustainability. They are made from renewable biological resources, which reduces reliance on non-renewable fossil-based materials
. By replacing conventional plastics with more sustainable bioplastics, biolaminates contribute to a greener manufacturing process and product lifecycle
-The article could benefit from a direct comparison between keratin biolaminates and other conventional insulating materials to clearly highlight the advantages of the new material.
It was added: A comparison of the thermal insulation properties of the tested composites with other types of biomaterials was demonstrated. This showed good thermal insulation properties of created biolaminate with keratine flour. E.g. thermal conductivity for wool is 0.0385–0.040 W/mK. For wood fiber insulation board has a thermal conductivity of 0.035 W/mK; Hemp products showed 0.05 W/mK; mussel shell (TC of 0.12 W/mK), or wine waste composite (TC of 0.35 W/mK) [Cosentino, L 2023].
Despite the focus is on the reuse of waste, no quantitative data is presented on the impact on the environment of the production and use of biolaminates that could strengthen the argument regarding environmental benefits.
It was added (page 2) ).
In the article [Zhang L. 2023] describes the threats resulting from the constant increase in poultry production. Feather waste is byproduct that is massively generated by the poultry industry globally Bird feathers are mainly composed of keratin, which is an insoluble, fibrous, recalcitrant protein, and it accounts for 85% to 99% of the total dried feather weight. However, feather waste is also a serious concern that may lead to environmental pollution because a lot of pathogens and microorganisms such as Salmonella and Vibrio can be found in poultry feathers. Various pollutants, such as ammonia, nitrous oxide, and hydrogen sulfide, can also be emitted from feather waste, which has brought about threats and risks to both human health and environmental safety [Zhang L. et al. 2023, Karuppannan S.K. et al., 2020].
(page 20)
While the production of creatine and the use of biolaminates have environmental implications, there are promising strategies to mitigate these impacts. Biotechnological advancements, renewable energy, and regulatory measures can improve the sustainability of creatine production.
The valorization of wood waste into biopolymers and biocomposites represents a sustainable alternative to traditional plastics, further supporting the transition to a circular economy. Although it is need of further research and comparisons with conventional material and the assessment of environmental impact benefits.
-There is no research question and should be added.
It was added: The aim of the research was to determine the effect of the relative humidity of the environment and the mass of moisture supplied to the bottom layer of the biolaminate on its values ​​of thermal resistance and thermal conductivity.
-There is no particularly well-founded literature review and should be considered.
It was added new literature:
Cetiner, I.; Shea, A.D. Wood Waste as an Alternative Thermal Insulation for Buildings. Energy Build. 2018, 168, 374–384
Cosentino, L.; Fernandes, J.; Mateus, R. A Review of Natural Bio-Based Insulation Materials. Energies 2023, 16, 4676. https://doi.org/10.3390/en16124676
Zhang, Long, Jingzheng Ren, and Wuliyasu Bai. 2023. "A Review of Poultry Waste-to-Wealth: Technological Progress, Modeling and Simulation Studies, and Economic- Environmental and Social Sustainability" Sustainability 15, no. 7: 5620. https://doi.org/10.3390/su15075620
Karuppannan, S.K.; Dowlath, M.J.H.; Raiyaan, G.I.D.; Rajadesingu, S.; Arunachalam, K.D. Application of poultry industrywaste in producing value-addedproducts—A review. In Concepts of Advanced Zero Waste Tools: Present and Emerging Waste Management Practices; Hussain, C.M., Ed.; Elsevier Inc.: Amsterdam, The Netherlands, 2020; pp. 91–121.
Comparisons with conventional material :
It was added:
A comparison of the thermal insulation properties of the tested composites with other types of biomaterials was demonstrated. This showed good thermal insulation properties of created biolaminate with keratine flour. E.g. thermal conductivity for wool is 0.0385–0.040 W/mK. For wood fiber insulation board has a thermal conductivity of 0.035 W/mK; Hemp products showed 0.05 W/mK; mussel shell (TC of 0.12 W/mK), or wine waste composite (TC of 0.35 W/mK) [Cosentino, L 2023].
Best regards,
Author
Reviewer 3 Report
Comments and Suggestions for Authors
After review the manuscript, the reviewer wish to thanks to authors for corrections done, now the manuscript shows significant improve, but still there are some points that need to be corrected,
- Keratin is the main component of birds feathers, and in some parts of discussion indicate keratin and in others still refer to bird feathers, please correct this.
- Table 2 it looks too crowded in the way which is reported, may be due the line numbers are overwriten with table text.
- In figure 3 and 4, In figures indicate Ceratine content, but caption is wirtenn creatine, which is correct?
- the subsections in Results section must be numerated.
- Conclusions must be spliced from discussion, discussion can be inserted in Results section, and Conclusions must be one individual section.
-Please follow Instructions for authors to report and insert in main text, the references.
Author Response

(The authors gave the same response as above.)
